# Testing the Intermediary Role of Perceived Stress in the Relationship between Mindfulness and Burnout Subtypes in a Large Sample of Spanish University Students

**DOI:** 10.3390/ijerph17197013

**Published:** 2020-09-25

**Authors:** David Martínez-Rubio, Juan P. Sanabria-Mazo, Albert Feliu-Soler, Ariadna Colomer-Carbonell, Cristina Martínez-Brotóns, Silvia Solé, Cristina Escamilla, Elisa Giménez-Fita, Yolanda Moreno, Adrián Pérez-Aranda, Juan V. Luciano, Jesús Montero-Marín

**Affiliations:** 1Psicoforma Integral Psychology Center, 46001 Valencia, Spain; david@psicoforma.es (D.M.-R.); cristina@psicoforma.es (C.M.-B.); 2Excellence Research Network PROMOSAM (PSI2014-56303-REDT), 28029 Madrid, Spain; 3Obesity Unit. QuironSalud, 46010 Valencia, Spain; 4Institut de Recerca Sant Joan de Déu, 08950 Esplugues de Llobregat, Spain; jp.sanabria@pssjd.org (J.P.S.-M.); jvluciano@pssjd.org (J.V.L.); 5Teaching, Research & Innovation Unit, Parc Sanitari Sant Joan de Déu, 08830 St. Boi de Llobregat, Spain; 6Faculty of Psychology, Universitat Autònoma de Barcelona, 08193 Bellaterra (Cerdanyola del Vallès), Barcelona, Spain; adrian.perez@uab.cat; 7Department of Medicine, International University of Catalonia, C/Josep Trueta s/n, 08195 Sant Cugat del Vallès, Barcelona, Spain; 8Department of Nursery and Physiotherapy, Universitat de Lleida, 25198 Lleida, Spain; silviasole@infermeria.udl.cat; 9Department of Psychology, Faculty of Health Sciences, European University of Valencia, 46010 Valencia, Spain; mariacristina.escamilla@universidadeuropea.es (C.E.); elisa.gimenez@universidadeuropea.es (E.G.-F.); 10Department of Basic Sciences, Faculty of Physical Activity and Sport Sciences, Catholic University of Valencia San Vicente Mártir, 46001 Valencia, Spain; yolanda.moreno@ucv.es; 11Department of Psychiatry, University of Oxford, Warneford Hospital, Oxford OX3 7JX, UK; jesus.monteromarin@psych.ox.ac.uk

**Keywords:** burnout, mindfulness, stress, mental health, students, university, structural equation model, cross-sectional study

## Abstract

The burnout syndrome is the consequence of chronic stress that overwhelms an individual’s resources to cope with occupational or academic demands. Frenetic, under-challenged, and worn-out are different burnout subtypes. Mindfulness has been recognized to reduce stress, comprising five facets (observing, describing, acting with awareness, non-judging of inner experience, and non-reactivity to inner experience). This cross-sectional study aimed to assess the relationship between mindfulness facets, perceived stress, and burnout subtypes in a sample of 1233 students of Education, Nursing, and Psychology degrees from different universities of Valencia (Spain). Structural Equation Modelling (SEM) was computed showing an adequate fit (Chi-square, CFI, TLI, RMSEA, and SRMR). Four mindfulness facets (all but observing) significantly correlated with general second-order mindfulness. Unexpected results were found: Acting with awareness facet was positively associated with frenetic subtype, while the non-reacting facet was positively associated with frenetic and under-challenged subtype. Ultimately, mindfulness facets negatively predicted the perceived stress levels, which in turn, predicted burnout. However, mindfulness plays different roles in the early stages of burnout syndrome (i.e., frenetic and under-challenged).

## 1. Introduction

### 1.1. Stress Factor and Burnout

When environmental demands overwhelm an individual’s resources, high levels of stress are perceived and threaten personal wellbeing [1,2]. Undergraduate and graduate students consistently experience elevated perceived stress due to academic, social, and financial pressures. In this regard, 50.8% of undergraduate students reported being “often” or “always” stressed [3]. Elevated levels of stress have a negative effect on mental and physical health [4]. A survey of undergraduate and graduate students found this factor to be the most usually reported impediment to academic performance [5]. Furthermore, stress is positively correlated with depression, a mental disorder that can also hamper academic performance [6]. Considering this background, student distress, along with associated anxiety and depression, has been increasingly recognized as a public health issue in various countries (e.g., [7]).

As a consequence of chronic perceived stress and highly related to poorer health [8,9], burnout syndrome is the inability to cope with chronic occupational stress and is an attempt to adapt to or protect oneself from it [10]. It is an adverse experience, comprising cognitions, emotions, and behaviors towards work, other people interacting with the individual at the workplace, and the professional role [11]. Burnout was originally conceptualized in helping professions, such as medical and other health-care professions [12]. However, it has also been observed in different types of professions and occupations [13], and also in graduate and undergraduate students. Even though students do not receive any salary, they are engaged in a structured and compulsory activities such as attending classes and completing academic assignments, which can be seen as a form of “work”; furthermore, they also experience high levels of pressure related to their studies and overwork [14,15]. Hence, students can also develop burnout [16,17,18,19,20,21]. In this sense, some systematic reviews have reported a prevalence of academic burnout in about 50% of university students [7].

Burnout syndrome comprises a state of emotional exhaustion, depersonalization, and a decreased sense of accomplishment when high levels of stress are perceived without the environmental resources needed to face the situation [22]. Exhaustion is the feeling of not being able to offer any more of oneself at an emotional level; depersonalization represents a distant attitude towards work; and decreased sense of accomplishment is the feeling of not performing tasks adequately or being incompetent. A more comprehensive definition of burnout has recently been proposed to differentiate three different clinical subtypes of the burnout syndrome [2]: (1) *Frenetic*, which is characterized by overload and the perception of jeopardizing one’s health to pursue worthwhile results, and is highly associated with exhaustion; (2) *under-challenged*, which is characterized by lack of development, defined as the perception of a lack of personal growth, together with the desire for a more rewarding occupation that better corresponds to one’s abilities, and is most strongly associated with cynicism; and (3) *worn-out*, which is characterized by neglect, defined as an inattentive and careless response to responsibilities, and is closely associated with inefficacy [23].

This typology emphasizes a phenomenological perspective and makes it possible to address the specific sources of subjective suffering [24], grouping distinct degrees of effort and persistence together in the same framework [25,26], which is rooted in a differential pattern of coping styles [27,28]. In this way, the burnout profiles can be ordered according to level of dedication to tasks, which affects the way individuals manage their feelings of distress. Altering the level of dedication to tasks may be a way for individuals to exert some control over the balance between efforts and rewards [24]. This classification criterion is consistent with the idea of a developmental transition between the different burnout subtypes driven by changes in dedication and coping style (from active to passive), from more (i.e., frenetic) to less (i.e., worn-out) dedicated. Each stage of burnout also corresponds to a different pattern of perceived stress as a result of differing levels of dedication [2,23,26,29,30]. It has been observed that the frenetic burnout subtype is associated with a coping style focused on active problem solving; the under-challenged burnout subtype presents an escapist coping style based on distraction and cognitive avoidance; while the worn-out burnout subtype is related to a coping style mainly characterized by behavioral disengagement [27].

### 1.2. Mindfulness

In recent years, there has been increasing evidence that acknowledges the potential role of mindfulness in promoting wellbeing and reducing anxiety, depression, and academic stress across a variety of populations [31], including graduate and undergraduate students [32,33,34,35,36,37]. Mindfulness is defined as the awareness that emerges through paying attention on purpose, in the present moment, and nonjudgmentally to the unfolding of experience moment by moment [38]. This particular quality of awareness has been associated with several indicators of mental and physical health and can be developed using specific techniques (mindfulness practices) commonly delivered in mindfulness-based interventions.

Mindfulness was initially considered as a one-dimensional construct, with tests yielding a single score [39]. Nonetheless, recently, a more complex conceptualization of mindfulness emerged and has been mostly adopted in research [40]. This conceptualization proposes the existence of five distinct facets empirically derived: (1) *observing* (noticing internal/external experiences), (2) *describing* (labelling experiences with words), (3) *acting with awareness* (focusing attention on one’s current activity), (4) *non-judging of inner experience* (experiencing thoughts/feelings without judging them or criticizing oneself), and (5) *non-reacting to inner experience* (allowing thoughts/feelings to come and go without reacting to them or getting caught up in them).

Overall, dispositional mindfulness has been found to act as a positive skill that fosters mental health [41,42] and, in the academic field, has been found in several studies to correlate with psychological wellbeing and mental health (e.g., [35,43]). To this extent, in the validation study of the Five Facet Mindfulness Questionnaire, Baer et al. [40] examined associations between mindfulness facets and related constructs in a sample of 881 American students, finding that three of the facets (acting with awareness, non-judging, and non-reacting) significantly predicted better general mental health; on the other hand, the observing facet was not clearly associated with fewer psychological symptoms. Harrington, Loggredo, and Perz [44] found an association between acting with awareness, describing, and non-judging facets and psychological wellbeing in 184 undergraduate American students. Bergin and Pakenham studied a sample of 481 Australian law school students and discovered that dispositional mindfulness moderated the effect of perceived academic stress on anxiety and depression [45]; furthermore, results indicated that higher levels of mindfulness facets (especially non-judging and acting with awareness) were related to better psychological adjustment. However, in contrast to predictions, observing was positively related to anxiety.

In another study conducted with 310 American undergraduates, Bodenlos and colleagues [46] highlight that mindfulness facets differently correlated with wellbeing; in this case, the observing facet was negatively associated with physical health, and both acting with awareness and non-judging facets were positively associated with emotional wellbeing; the latter facet also correlated positively with social functioning. Results suggest that mindfulness (understood as a set of cognitive skills) could act as a personal internal resource that buffers against mental health outcomes such as academic burnout [47,48,49] by increasing individuals’ ability to detect the signs of academic stress through the improvement of awareness [44] and, in turn, increases access to coping resources and helps alleviate the negative effects of academic stress [50].

Given perceived academic stress was one of the significant predictors of burnout [2], cultivating mindfulness may reduce burnout via alleviating psychological academic distress. Mindfulness facets have been found to explain the three components of burnout (exhaustion, cynicism, and inefficacy) in a study with 381 employees, even after controlling for other significant predictors such as personal traits, workplace resources, and workplace demands. More precisely, non-judging and non-reacting mindfulness facets were shown to convey the relationship between mindfulness with burnout domains [51]. Research has also explored the mediating role of stress in mindfulness and health behaviors. At the same time, a study conducted with 385 college students found that the relationship between mindfulness and health behaviors (e.g., sleep quality, physical activity, activity restriction, and binge eating) was partially mediated by stress [52]. In the same line, in another study with 427 occupational therapists, stress mediated the relationship between mindfulness components and health behaviors [53]. Similarly, a cross-sectional study of 315 soldiers identified that perceived stress fully mediated the effects of unidimensional mindfulness on mental health [54].

However, there are not many studies assessing the relationship between mindfulness facets, perceived academic stress, and burnout subtypes in university students, nor that evaluate the explanatory power of the mindfulness construct regarding the burnout subtypes. Knowing how mindfulness specific facets interact with perceived academic stress and burnout subtypes could offer clues to promote programs fostering such cognitive skills in academic settings. The possibly different connection between the mindfulness facets and each burnout subtype might provide useful information for developing programs targeting each specific profile, efficiently administered through brief [55] or online interventions [56].

### 1.3. Aims and Hypotheses

In this sense, the aim of this study was to assess the relationship between mindfulness facets, perceived stress, and burnout subtypes in a sample of 1233 students of Education, Nursing, and Psychology degrees from different universities of Valencia (Spain). We started with the following assumptions: All mindfulness facets but observing will be directly and negatively related with burnout dimensions (overload, lack of development, and neglect), and indirectly through the intermediary latent factor of perceived academic stress (Hypothesis 1). We also expect the mediating role of perceived academic stress to be greater in the case of overload (active coping style) and lesser in neglect (passive coping style), with intermediate levels in the lack of development (which is more prone to cognitive avoidance and shows intermediate levels of dedication) (Hypothesis 2) [27].

## 2. Materials and Methods

### 2.1. Research Design

Cross-sectional study with a convenience sample of Spanish university students. Participants were included if: (1) Aged ≥18 years, (2) college students in the field of Education, Nursing, or Psychology, (3) signed informed consent, and (4) able to understand Spanish.

### 2.2. Participants

Study participants were Spanish university students from Education (*n* = 589, 47.8%), Nursing (*n* = 385, 31.2%), and Psychology (*n* = 259, 21%) degrees (*n* = 1233) from different universities of Valencia (Spain). They were in their first (*n* = 455, 36.9%), second (*n* = 296, 24%), third (*n* = 195, 15.8%), fourth (*n* = 237, 19.2%), or fifth (*n* = 49, 4%) academic course. Most of the participants were women (78.8%) between the ages of 18 and 56 (*M* = 22.19, *SD* = 5.18). More details can be found in Table 1. Available sample size was considered in according to the recommended 10:1 ratio of the number of participants to the number of the test items [57].

### 2.3. Instruments

#### 2.3.1. Sociodemographic Data

The following sociodemographic characteristics of the sample were collected: Age, gender, year of study (“first”, “second”, “third”, “fourth”, “fifth”), university degree, academic year, stable relationship (“yes”, “no”), with a stable job (“yes”, “no”), living arrangement (“with family”, “student residence”, “shared flat”, “living alone”, “living as a couple”), with a scholarship grant (“yes”, “no”).

#### 2.3.2. Measures

The Spanish version of The Burnout Clinical Subtype Questionnaire, short form (BCSQ-12; [26]) is a 12-item questionnaire comprising three dimensions: (a) Overload” (e.g., “I think I invest more than is healthy in my commitment to my studies”), (b) lack of development (e.g., “I would like to study something else that would be more challenging to my abilities”), and (c) neglect (e.g., “When the results of my studies are not good at all, I stop making an effort”) in a 7-point Likert-type scale (from 1 = “completely disagree” to 7 = “completely agree”). The BCSQ-12 demonstrated high internal consistency for the three subscales in the present sample with Cronbach’s α of 0.81 (overload), 0.73 (lack of development), and 0.84 (neglect).

The Spanish version of the Perceived Stress Questionnaire is a 24-item (PSQ-24) that assesses the subjective experience of perceived stressful situations and stress reactions during the last 30 days [58,59]. The participants indicate their agreement with the items in a 4-point Likert scale (from 1 = “almost never” to 4 = “almost always”). This PSQ-24 was validated in a university student sample and is able to measure “global” perceived stress and two stress-related dimensions [60]: (1) “tenseness” or the perception of hurry and overdemand (e.g., “You have too many things to do”), and (2) “frustration” or the negative affective response to stress including feelings of discouragement, joylessness, and worry (e.g., “You feel discouraged”). A Cronbach’s alpha of 0.91 for the global perceived stress scale, and 0.88 for tenseness, and 0.87 for frustration subscales were found in the present study sample, indicating good internal consistency.

The Five Mindfulness Facets Questionnaire, 20-item short form (FFMQ-20; [61]) is a measure with 20 items assessing the five facets of mindfulness: Observing, describing, acting with awareness, non-judging of inner experience, and non-reactivity to inner experience. The response format is in a 5-point Likert scale (from 1 = “never or very rarely true” to 5 = “very often or always true”). The FFMQ-20 has been found to be valid and reliable to measure the experience of mindfulness [62]. Since the observing facet has shown inconsistent results in non-meditative samples, this specific mindfulness facet was not included in the present study. The internal consistency (Cronbach’s αs) of the FFMQ subscales were globally satisfactory in this sample: 0.70 (describing), 0.83 (acting with awareness), 0.81 (non-judging), and 0.57 (non-reacting).

### 2.4. Procedure

Students were asked to voluntarily respond to the study questionnaire through an online survey (https://es.surveymonkey.com) or via paper-and-pencil format. Participants were provided with a general overview of the aims and characteristics of the study and an informed consent was obtained from all participants before they fulfilled the survey. The study was developed according to national and international standards (Helsinki and Tokyo Convention) and was approved by the Research Ethics Committee of the University of Valencia (code H1455835241950). Data were treated anonymously and were only used for the purposes of the study and thus the confidentiality of the participants included in the study was guaranteed. The study survey was made available from December 2015 to May 2016.

### 2.5. Data Analyses

Sociodemographic data were analyzed with descriptive statistics of mean (*M*), standard deviation (*SD*), and range. Analyses were conducted with SPSS v25.0 (SPSS Inc., Chicago, IL, USA) and the MPLUS v7 packages (Múthen & Múthen, Los Angeles, CA, USA). All the tests used were bilateral and the significance level was set at α < 0.05.

Structural Equation Modelling (SEM) procedure was used to investigate the impact of mindfulness facets on burnout subtypes and the intermediary role of perceived stress. SEM model was created by considering the four mindfulness facets (all but observing) with solid evidence of positive effects on mental health in non-meditative samples [63]. The four mindfulness facets were taken as exogenous variables in SEM model, global perceived stress as endogenous variable, and burnout subtypes as final outcomes. SEM analysis provides information about consistency of the hypothesized mediational model to the data. Measurement models were first tested to assess whether each latent variable was represented by its indicators. If the measurement models turned out satisfactory, then the SEM was tested. We tested the following measurement models: (a) A four-facet (without observing) hierarchical model of the FFMQ since it is recommended to exclude the observing facet in non-meditative samples [63], which provides a comprehensive view of mindfulness skills [35] and has proven its better fit (compared to the five-facet hierarchical model) in general population (e.g., [64]); (b) the bifactor model of the PSQ with two sub-factors (‘frustration’ and ‘tenseness’) with regard to one general factor (‘global perceived stress’) since it was reported to be satisfactory in a Spanish dental student sample [60]; and (c) a three-factor correlated model of the BCSQ since it showed a good fit in previous study with a Spanish sample [28].

After confirming goodness of model fit for the three scales, a SEM was computed by considering the four mindfulness facets as independent variables, global perceived stress as intermediary variable, and burnout subtypes as dependent variables. Given that factor loadings and standard errors are underestimated when using maximum likelihood estimation with categorical variables [65], we applied mean- and variance-corrected weighted least squares (WLSMV) to test the fit of the factor structures of the scales and as well as for the structural equation models. WLSMV estimation uses the polychoric correlation matrix of the items, it is suited for ordered categorical variables and provides robust parameter estimates, standard errors, and tests of model fit [66]. In addition to the chi-square test, the following fit indices were analyzed (the values in parentheses denote goodness-of-fit standards in accordance with Schermelleh-Engel et al. [67] using conservative and liberal cut-offs): Comparative Fit Index (CFI) and Tucker-Lewis Index (TLI; ≥0.95 or ≥0.90), Root Mean Square Error of Approximation (RMSEA) with 90% confidence intervals (≤0.06 or ≤0.08), and Standardized Root Mean Squared Residual (SRMR). Since only 39 participants had incomplete data (3.2% of the total sample), no procedure for handling missing data was applied.

## 3. Results

### 3.1. Correlations of the Model Variables

As shown in Table 2, all mindfulness facets variables showed significant correlations with burnout subtypes. Mainly, overload indicated a positive correlation with perceived stress facet (*r* = 0.48) and a negative with non-judging facet (*r* = −0.17); lack of development a positive with perceived stress facet (*r* = 0.18) and a negative with awareness facet (*r* = −0.14); and neglect a negative with awareness (*r* = −0.34), non-judging (*r* = −0.22), describing (*r* = −0.18), and non-reacting (*r* = −0.11) facets. The descriptive analyses of all the variables are presented in Table 1.

### 3.2. Measurement Models

All factorial loadings for items in each FFMQ subscale were significant (with λ from 0.195 (#18; describing facet) to 0.851 (#16; non-judging facet)). Furthermore, the four mindfulness facets significantly correlated with a general second-order mindfulness factor (from 0.295 (non-reacting facet) to 0.707 (non-judging)). Factorial loadings between items in PSQ and a general perceived stress factor were all significant and in the expected direction (with λ from 0.271 (#7) to 0.725 (#26)). Factor loadings between items and with each BCSQ subscale were also significant and showed acceptable values (with λ from 0.267 (#2) to 0.843 (#5)). According to modification indexes, correlations between the error terms of items #18 and #20 in FFMQ (describing subscale) and #2 and #8 in BCSQ (lack of development scale) were introduced in each corresponding model. The incorporation of these correlations between error terms significantly reduced the χ^2^ (*p* < 0.01). The measurement models were found to be satisfactory, showing fit indexes above standards cut-off (see Table 3 for more details).

### 3.3. Structural Equation Model for Mindfulness Facets Predicting Burnout Subtypes and the Intermediary Effect of Perceived Stress

Although chi-squared statistic was found to be significant (χ2 (1222) = 4925.267, *p* < 0.001), overall structural model showed an adequate fit according to liberal cut-offs (RMSEA[CI90%] = 0.050 [0.049, 0.052]; CFI = 0.909; TLI = 0.901; SRMR = 0.059). As can be seen in Figure 1, the following relationships between mindfulness facets and global perceived stress with burnout subtypes were found: Overload subtype was significantly (and positively) predicted by acting with awareness (γ = 0.202; *p* < 0.001), non-reacting (γ = 0.173; *p* < 0.001), and perceived stress (β = 0.714; *p* < 0.001); lack of development was positively predicted by non-reacting (γ = 0.180; p < 0.001) and perceived stress (β = 0.347; *p* < 0.001) and negatively predicted by act with awareness (γ = −0.185; *p* < 0.001); and neglect subtype was positively predicted by perceived stress (β = 0.176; *p* < 0.001) and negatively by acting with awareness (γ = −0.310; *p* < 0.001). Global perceived stress was significantly and negatively predicted (in descending order) by non-judging (γ = −0.316; *p* < 0.001), non-reacting (γ = −0.204; *p* < 0.001), acting with awareness (γ = −0.151; *p* < 0.001), and describing (γ = −0.108; *p* = 0.013). Burnout dimensions correlated significantly (overload-lack of development (r = 0.30, *p* < 0.001), overload-neglect (r = 0.091, *p* = 0.025), lack of development-neglect (r = −0.500, *p* < 0.001)).

The removal of non-significant relationships yielded a model with adequate fit and similar associations—compared to the first model—among latent variables (χ2(1229) = 4809.296, *p* < 0.001, RMSEA[CI90%] = 0.049 [0.048, 0.051]; CFI = 0.912; TLI = 0.905). The cases-to-free parameter ratio for our general model was 3.81 (1192/313). This ratio was below the 5:1 threshold suggested by Bentler and Chou [68]. However, as the effect of a low cases-to-parameter ratio on the estimates is less severe than having a small sample size [69], the size of our sample (*n* = 1192) probably pays off part of the negative consequences related to abundance of parameters in our model.

## 4. Discussion

In this first study exploring the relationship between mindfulness facets and burnout subtypes in university students whilst considering the intermediary effect of perceived academic stress, expected and unexpected associations between mindfulness and burnout subtypes have been found. In this sense, along with the anticipated negative predictive effect of mindfulness on perceived stress, unexpected *positive* associations were found for acting with awareness facet with overload dimension (corresponding to frenetic burnout subtype), and between non-reacting mindfulness facet with the same burnout dimension and with lack of development (corresponding to under-challenged burnout subtype). Acting with awareness also predicted—negatively, as expected—lack of development and neglect (corresponding to worn-out burnout subtype). Coherently with the literature, perceived academic stress was found to be positively associated with all burnout dimensions and in a decreasing manner from overload to neglect burnout dimensions, as proposed in the second hypothesis. Hence, the first hypothesis was only partially accomplished because not all mindfulness facets were directly and negatively associated with burnout dimensions, but all of them were indirectly related through the intermediary latent construct of perceived academic stress.

Our results showing positive associations between mindfulness facets and burnout are different from those obtained in a previous cross-sectional study in primary care physicians (not students) assessing the predictive effects of Mindfulness Awareness Attention Scale (MAAS, [39]) along with resilience on burnout (using also the BCSQ) [28]. In this study, a negative association (of small magnitude, γ = −0.25) between mindfulness and overload facet was reported. Differences between both studies are even more surprising when considering that acting with awareness facet from the FFMQ evaluates a very similar mindfulness construct to that assessed in MAAS. In fact, the FFMQ acting with awareness subscale, in its original version, was mainly based on a selection of items from the MAAS (i.e., 5 out of 8 items; [40]). However, it is also worth to mention that the 20-item version of the FFMQ [61] used in the present study only included one item from the MAAS (out of 4), with two more items from the Kentucky Inventory Mindfulness Skills [70] and one from the Cognitive and Affective Mindfulness Scale [71], so additional divergence from Montero-Marin’s study [28] could also be expected. Furthermore, Montero-Marin et al. [28] explored mindfulness–burnout relationships in a sample of healthcare professionals and not university students, so burnout constructs may not be totally overlapped with slightly different interpretation of the content of the scale depending on the sample. Perhaps these discrepancies between both studies are due to target population. The psychological construct of mindfulness applied to the perceived academic stress could exhibit idiosyncratic paths between mindfulness facets and burnout subtypes that are not explored in the health care professional setting. In the present research, the university students are probably in an early stage of burnout syndrome, more characterized by the overloading academic demands (i.e., a frenetic burnout subtype) in contrast to the Montero-Marin’s study [28] that focuses on professional health care. That is why acting with awareness could facilitate the consciousness of being in this early burnout stage. Additionally, we observed significant negative effects of acting with awareness on lack of development and neglect dimensions, which were not observed in Montero-Marin’s study [28], which suggests a potential protective effect of this mindfulness facet in under-challenged and worn-out burnout subtypes. Thus, our results confirm the second hypothesis in which we expected, in a gradual manner, the perceived academic stress role’ be modulated regarding its burnout levels (i.e., from more to less dedicated). The mediating role of perceived academic stress is greater in the case of frenetic subtype given its active coping style and lesser in the worn-out subtype due to its passive coping style. Furthermore, intermediate levels in the under-challenged subtype are reported [27].

In order to deeply explore these specific relationships among burnout subtypes and mindfulness facets, each mindfulness dimension will be described in detail. Acting with awareness concerns a sense of bringing awareness and continuous attention to the present moment and one’s behaviors (contrarily to living on “automatic pilot”) [72] and reflects both an implicit orientation of acceptance toward experience (being fully aware of present experience with openness) and attention monitoring of own actions [73]. Regarding the role of this facet in mindfulness, many mindfulness instructors propose the training of attention and awareness as foundational in mindfulness interventions aimed both at reducing academic stress in university settings and fostering resilience [33,74]. This mindfulness facet, along with non-judging, usually shows high correlations with positive and negative mental health outcomes such as satisfaction with life, perceived academic stress, anxiety, or depression [75]. Furthermore, in a study by Yang et al. [76] conducted in mental health professionals, acting with awareness presented the strongest association with stress and burnout in comparison to the other FFMQ facets. Continuous attention and high involvement in present-moment activities, which are reflected in acting with awareness facet (e.g., “When I do things, my mind wanders off and I’m easily distracted”—reversed scoring), seem to theoretically overlap with high levels of engagement that are inherent to frenetic burnout subtype and its corresponding overload dimension (e.g., “I overlook my own needs to fulfil studies demands”). Individuals classified in the *frenetic* subtype work (or study) to the point of exhaustion in search of good results, are highly involved in their work and invest great effort, usually becoming overloaded and putting in risk their own health and personal life in the pursuit of success [30].

Non-reacting refers to the tendency to allow internal experiences (thoughts and feelings) to come and go, without being caught up in or carried away by them (e.g., “I watch my feelings without getting lost in them.”) [72]. Regarding its positive association with overload dimension, high levels of non-reacting (“I watch my feelings without getting lost in them.”) could reflect a reduced cognitive elaboration when facing a difficult situation in academic settings. Such coping style could also be characteristic of frenetic subtype since a coping style focused on actively solving situations have been found to be relevant in this burnout profile [27], explaining—at least partially—our findings regarding this mindfulness facet. Furthermore, in our study, higher scores of non-reacting (e.g., “In difficult situations, I can pause without immediately reacting”) predicted greater levels of lack of development (e.g., “My studies don’t offer me opportunities to develop my abilities”) in academical perform. The negative association between non-reacting and perceived academic stress along with previous literature supporting the positive effect of this mindfulness facet on mental wellbeing (e.g., [53,75,77]), it seems reasonable that the unexpected pattern observed regarding lack of development could rely on a potential misunderstanding of its items in our student sample and not on a harmful effect of this mindfulness facet on burnout. In this line, taking a non-reactive stance regarding internal events (i.e., non-reactivity or equanimity) and indifference or passivity (which are inherent to under-challenged burnout subtype) are concepts that classically have been mistakenly mixed, to the point that they are even considered “near-enemies” [78]. In contrast to the concept of equanimity, which includes an orientation to care and attentiveness to experience, students scoring high in lack of development (i.e., under-challenged burnout subtype) would display a lack of engagement with their student activities, which are perceived as unstimulating and self-limiting, leading to carry out tasks with indifference [30].

At the same time, regarding our findings with the non-reactivity facet, we cannot rule out a potential measurement effect partially explaining our results. In this regard, in non-meditative samples, studies have reported that non-reacting mindfulness facet was a weak indicator of its intended construct [61] with weak measurement properties and a small association with overall mindfulness [40]. Furthermore, it is also well-known that shorter versions of scales (as it is the case for FFMQ and BCSQ versions used here) may tend to show poorer reliability and validity [79,80]. In this line, a far-from-good internal reliability in non-reacting scale was found in our sample (α~0.60).

In respect to neglect dimension (corresponding to worn-out burnout subtype), it is noteworthy that our model confirms that this is the academic burnout subtype with less associated perceived academic stress, probably due to a total lack of involvement in their studies [60]. Individuals presenting this type of burnout give up when faced with academic stress or absence of reward (e.g., “When the effort I invest in work is not enough, I give in”). They usually show feelings of lack of control and lack of acknowledgment for their tasks, leading them to cope passively and to neglect their responsibilities [30]. Acting with awareness facet negatively predicted neglect. These results may suggest that increasing awareness could be particularly therapeutic in this burnout profile, perhaps promoting a more positive orientation to experience and/or increasing behavioral engagement, which lacks in this burnout subtype [27]. At the same time, this negative relationship seems to mirror (inversely) the association between acting with awareness and overload dimension. Since overload and neglect represent extremes in effort/dedication continuum in burnout subtypes model [30], acting with awareness could also be inversely related to neglect because scoring high in this mindfulness facet may entail higher involvement in work/studies.

Interestingly, non-judging was the mindfulness facet with a greater explanatory power of perceived academic stress. However, it did not present any direct association with any burnout dimension. Since burnout is a response to the failure to cope adequately with chronic academic/occupational stress and is an attempt to adapt from it [12], it seems reasonable to suggest that non-judging could also have a beneficial effect on students with burnout through reducing the levels of academic stress. Additionally, there is plenty of evidence pointing out that chronic stress impacts on quality of life and general health (e.g., [81,82]) so promoting a non-evaluative stance toward thoughts and feelings deems interest for promoting wellbeing in academicals settings [72]. Finally, the describing facet was not directly related to burnout dimensions and a very small predictive effect on perceived academic stress was reported (γ = −0.108). Describing refers to the ability of labelling internal experiences with words [72] and it has been stated that describing is not central to the construct of mindfulness [83], as it involves additional cognitive processing and elaboration beyond monitoring one’s experiences (which may also help in emotion regulation). Coherently with our results, modest effects (especially when compared to those with non-judging and acting with awareness facets) on mental health and wellbeing have been reported in university students [75].

The prevalence of mental health problems and burnout in university students is a current challenge in the education field. In light of our findings, further tailored interventions for burnout prevention in academicals settings should contemplate burnout subtypes as potential indicators for adapting mindfulness teachings and other psychotherapeutic approaches to obtain greater benefits. In this regard, current psychotherapeutic recommendations for burnout subtypes [84] may also take into consideration our findings (particularly on acting with awareness and non-reacting facets with some burnout dimensions). Our findings can provide some insights for further adaption of mindfulness exercises and teachings in mindfulness interventions provided to university students (for example, stressing that *non-reacting is not indifference* in students within the under-challenged subtype). Combining the identification of burnout subtypes and tailoring mindfulness contents and practices accordingly may increase the efficacy of mindfulness-based interventions in students, a population for which, to date, there is still much room for improvement [85].

Moreover, our results suggest that mindfulness may play a different role depending on the burnout stage; mindfulness-based interventions could be more beneficial for later burnout stages than for the earlier ones—if ordered as follows: Frenetic, under-challenged, and worn-out. Unexpected results regarding acting with awareness and non-reacting mindfulness facets point out the need for incorporating cognitive elements to mindfulness when addressing early burnout stages to accommodate their coping styles and to better disentangle the lack of development feelings from the non-reacting mindfulness facet. Targeting specific mindfulness programs depending on its specific burnout stages open the door to a more complex and precise conceptualization and in turn, its psychotherapy intervention. Given the broad range of benefits of mindfulness [32,33,36,49] to reduce perceived academic stress and prevent, in turn, burnout syndrome, psychological counselling at the university setting could be an important aspect to point out. University students face cognitive and emotional challenges due to their studies’ development with very few academic resources to deal with their own situation. Being aware of the role of the mindfulness facets to each burnout subtype will allow us to better provide specific trainings needed to deal with their academic demands and psychosocial pressure in a way adjusted to the characteristics of each particular case.

We must acknowledge some limitations in the present study. First, we employed a non-representative convenience sample of Spanish students, and thus, our findings may not be completely generalizable to the population. Second, our data are cross-sectional, so conclusions about the direction of the associations between the variables cannot be drawn. Future longitudinal studies should evaluate the effects the effects of mindfulness on burnout subtypes and the intermediary effect of perceived academic stress. In order to deeply explore the relationship among mindfulness facets and burnout subtypes, future research should explore how specific mindfulness-related exercises/trainings may have a differential effect in each burnout subtype. In this regard, we propose that more cognitive accompaniment should be required in the frenetic burnout subtype. Hence, our results show how mindfulness-based interventions might be insufficient in some burnout profiles pointing out the need of specific complementary cognitive psychotherapies. Some clinical implications an intervention targeting the different burnout subtypes should incorporate have already been proposed theoretically in a previous work including several therapeutic perspectives, as we also suggest here [84,86].

## 5. Conclusions

Study hypotheses were partially supported by our findings. All mindfulness facets negatively predicted levels of perceived stress, which in turn predicted burnout. However, only two (out of four) facets (i.e., acting with awareness and non-reacting) were directly related to burnout dimensions. These two facets showed different patterns of relationship with burnout, with acting with awareness having a dual effect on overload (positive), and on lack of development and neglect (negative); non-reacting mindfulness facet showed a direct positive predictive effect on the two first burnout dimensions. Our findings should be understood in the context of designing tailored psychotherapeutic programs (including mindfulness interventions) considering specific burnout subtypes.

## Figures and Tables

**Figure 1 ijerph-17-07013-f001:**
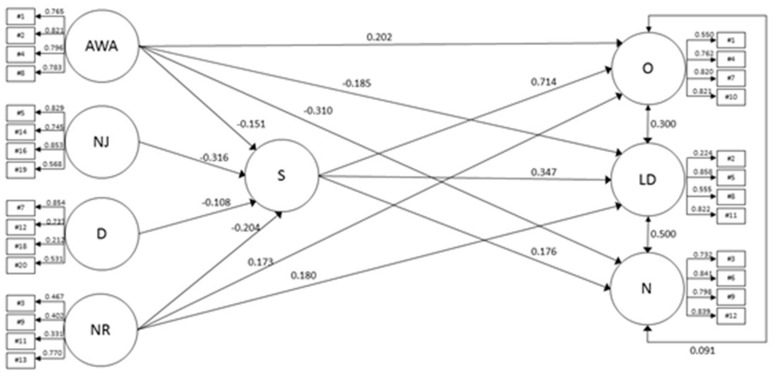
Structural equation model for mindfulness facets, perceived stress dimensions, and burnout subtypes. Note. Circles represent latent constructs and rectangles observable variables. Standardized factor weights are over the arrows. AWA = Acting with awareness; D = Describing; LD = Lack of development; O = Overload; N = Neglect; NJ = Non-judging; NR = Non-reacting. For clarity, non-significant pathways between latent variables were not shown. Factor loadings of items on general perceived stress factor (with *λ*s ranging from 0.185 [#4] to 0.768 [#20]; all *p* < 0.001), correlations between mindfulness facets and a second-order mindfulness factor (*rAWA*= 0.621, *rNJ*= 0.707, *rD*= 0.648, *rNR*= 0.314; all *p* < 0.001), and correlation between error terms between items #18 and #20 within D (*r*= 0.400, *p* < 0.001) and between items #2 and #8 within LD (*r*= 0.562, *p* < 0.001) were not shown for the same reason.

**Table 1 ijerph-17-07013-t001:** Sociodemographic and outcome variables.

*Sociodemographic*	Sample (*n* = 1233)
Age, *M (SD)*	22.19 (5.18)
Women, *n* (%)	971 (78.8)
Living arrangement, *n* (%)	
*Familiar residence*	907 (73.6)
*Student residence*	20 (1.6)
*Shared flat*	176 (14.3)
*Living alone*	38 (3.1)
*Living as a couple*	89 (7.2)
With a stable job, *n* (%)	188 (15.2)
University degree, *n* (%)	
*Psychology*	259 (21.0)
*Nursery*	385 (31.2)
*Education*	589 (47.8)
With a student grant, *n (%)*	309 (25.1)
Academic years, *M (SD)*	2.30 (1.26)
**Mindfulness facets (FFMQ-20)**	
*Acting with awareness* *M (SD)*	12.57 (3.43)
*Non-judging* *M (SD)*	13.75 (3.72)
*Observing* *M (SD)*	12.66 (3.63)
*Describing* *M (SD)*	13.15 (3.27)
*Non-reacting* *M (SD)*	11.81 (2.69)
**Perceived Stress Questionnaire (PSQ-24)** **Burnout subtypes (BCSQ-12)**	0.43 (0.17)
*Overload* *M (SD)*	3.13 (1.29)
*Lack of development* *M (SD)*	2.67 (1.15)
*Neglect* *M (SD)*	2.21 (1.02)

Note: BCSQ = Burnout Clinical Subtype Questionnaire; FFMQ = Five Mindfulness Facets Questionnaire.

**Table 2 ijerph-17-07013-t002:** Pearson correlations among all variables in the model.

	1	2	3	4	5	6	7	8
1. Awareness	-	-	-	-	-	-	-	-
2. Non-judging	0.362 *	-	-	-	-	-	-	-
3. Describing	0.259 *	0.253 *	-	-	-	-	-	-
4. Non-reacting	0.091 *	0.128 *	0.176 *	-	-	-	-	-
5. Perceived stress	0.292 *	−0.361 *	−0.229 *	−0.166 *	-	-	-	-
6. Overload	−0.013	−0.168 *	−0.078 *	−0.003	0.482 *	-	-	-
7. Lack of development	−0.143 *	−0.090 *	−0.002	−0.059 *	0.185 *	0.230 *	-	-
8. Neglect	−0.338 *	−0.217 *	−0.176 *	−0.113 *	0.245 *	0.112 *	0.311 *	-

Note: * *p* < 0.001.

**Table 3 ijerph-17-07013-t003:** Fit statistics for latent structure models for Five Mindfulness Facets Questionnaire (FFMQ-20), Perceived Stress Questionnaire (PSQ-24), Burnout Clinical Subtype Questionnaire (BCSQ-12), and SEM.

Model	χ^2^	df	RMSEA [90% CI]	CFI	TLI	SRMR
**FFMQ-20** Four-facet hierarchical model [63]	888.676 *	99	0.082 [0.077, 0.087]	0.931	0.917	0.064
**PSQ-24** Bifactor model [2]	1393.296	228	0.065 [0.061, 0.068]	0.949	0.938	0.047
**BCSQ-12** Three-facet correlated model [23]	374.508 *	50	0.073 [0.066, 0.080]	0.972	0.963	0.052
**SEM** FFMQ, PSQ & BCSQ	4925.267 *	1222	0.050 [0.049, 0.052]	0.909	0.901	0.059

Note: * *p* < 0.001. RMSEA: Root Mean Square Error of Approximation; CFI: Comparative Fit Index; TLI: Tucker-Lewis Index; SRMR: Standardized Root Mean Squared Residual; SEM: Structural Equation Modelling.

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
