# Peer review of "Testing the Intermediary Role of Perceived Stress in the Relationship between Mindfulness and Burnout Subtypes in a Large Sample of Spanish University Students"

_ijerph, 2020, doi:10.3390/ijerph17197013_

Round 1

Reviewer 1 Report

ABSTRACT
· In general it is correct. However, some aspects should be reviewed.
1. The APA regulation discourages listing citations from authors and specific results in the Abstract
2. Also insert the SRMR value as an element of statistical validity of the SEM model

1. Introduction
· It is suggested to establish two sections that order the review
1.1. Stress Factor and burnout
· As the study is carried out in the academic field, "academic stress" is also definitely suggested and not only "general": there are works in this Journal referring to this.
It is suggested to insert specific information on the types of emotional and coping strategies that favor burnout
1.2. Mindfulness training

1.3. Aims and hipotheses

2. Method
· It is suggested to reorder the sections in:
2.1 Participants
2.2 Instruments
2.3 Procedure
2.4 Design and Data analysis: it is suggested to insert the mean of the SRMR index

3. Results
· There is information in section 3.1. that is proper to the Participants. The information on the sociodemographic characteristics of the sample are not results. You must change places.
· Linear predictive results must be preceded by linear association results. Please insert the correlation results between all variables. In the same table they can include the descriptive results of the variables, mean (SD).
· The structural results are correct. It is suggested to insert the SRMR index

4. Discussion
· It is suggested to relate the discussion to the hypotheses (these must be more precise, for this).
· The discussion, in general, is correct, but it is suggested to make some reference to its contextualization in the university academic environment and in academic emotional stress. There are quite a few recent works on this.
· It is recommended to establish a section referring to the implications for psychological counseling at the University.

Reviewer 2 Report

The study is of scientific interest, providing useful information.

The introduction provides relevant information. It would be advisable to incorporate more updated bibliographies, especially in relation to studies on the university population.

Hypothesis 2 should be formulated more clearly indicating which variables are related to different patterns for each wear profile.

It is methodologically correct. The procedure is adequate, as well as the selected measuring instruments and the analyzes performed.

The results clearly show the data obtained.

The discussion reflects the most relevant information and establishes comparisons with other studies, proposing answers to the discrepancies found. They should incorporate more updated references

In conclusions, in addition to the usefulness of the data found and its applicability, they should indicate the future lines in which it would be necessary to deepen.

Reviewer 3 Report

This manuscript describes a cross-sectional study that aims to examine the relationships between stress, burnout, and mindfulness facets. Although the authors state that there are no studies on this topic, a quick Scopus search provided many similar studies, on the same topic, and with students. Therefore, the novelty of the study and its actual impact should be toned down. 

Regardless, the paper is well written and with a sound methodology. Considering the limited impact of the research question, however, it could be re-shaped as a research letter, and presents its results in one or two pages. 

Finally, in the abstract, introduction, and conclusions, there is some confusion about what mindfulness is: please disambiguate the fact that mindfulness is NOT a training, but a psychological construct. There are several mindfulness-based training options, including (but not limited to) meditation-based interventions.

Round 2

Reviewer 3 Report

I have no further comments